# Immune Evasion of SARS-CoV-2 Omicron Subvariants XBB.1.5, XBB.1.16 and EG.5.1 in a Cohort of Older Adults after ChAdOx1-S Vaccination and BA.4/5 Bivalent Booster

**DOI:** 10.3390/vaccines12020144

**Published:** 2024-01-30

**Authors:** Rafael Rahal Guaragna Machado, Érika Donizetti Candido, Andressa Simoes Aguiar, Vanessa Nascimento Chalup, Patricia Romão Sanches, Erick Gustavo Dorlass, Deyvid Emanuel Amgarten, João Renato Rebello Pinho, Edison Luiz Durigon, Danielle Bruna Leal Oliveira

**Affiliations:** 1Department of Microbiology, Institute of Biomedical Sciences, University of São Paulo, São Paulo 05508-000, SP, Brazil; 2Dom Pedro II Geriatric and Convalescent Hospital, Irmandade da Santa Casa de Misericórdia de São Paulo, São Paulo 02265-002, SP, Brazil; 3Department of Parasitology, Institute of Biomedical Sciences, University of São Paulo, São Paulo 05508-000, SP, Brazil; 4Hospital Israelita Albert Einstein, São Paulo 05652-900, SP, Brazil; 5Laboratório de Medicina Laboratorial (LIM03), Department of Pathology, School of Medicine, University of São Paulo, São Paulo 01246-903, SP, Brazil; 6Laboratório de Gastroenterologia Clínica e Experimental (LIM07), Department of Gastroenterology, School of Medicine, University of São Paulo, São Paulo 01246-903, SP, Brazil; 7Scientific Platform Pasteur-USP, São Paulo 05508-020, SP, Brazil

**Keywords:** SARS-CoV-2, Omicron sublineages, XBB.1.5, XBB.1.16, EG.5.1, ChAdOx1-S, Bivalent BA.4/BA.5 mRNA vaccine, neutralizing antibodies, immune escape, older adults

## Abstract

The recently emerged SARS-CoV-2 Omicron sublineages, including the BA.2-derived XBB.1.5 (Kraken), XBB.1.16 (Arcturus), and EG.5.1 (Eris), have accumulated several spike mutations that may increase immune escape, affecting vaccine effectiveness. Older adults are an understudied group at significantly increased risk of severe COVID-19. Here we report the neutralizing activities of 177 sera samples from 59 older adults, aged 62–97 years, 1 and 4 months after vaccination with a 4th dose of ChAdOx1-S (Oxford/AstraZeneca) and 3 months after a 5th dose of Comirnaty Bivalent Original/Omicron BA.4/BA.5 vaccine (Pfizer-BioNTech). The ChAdOx1-S vaccination-induced antibodies neutralized efficiently the ancestral D614G and BA.4/5 variants, but to a much lesser extent the XBB.1.5, XBB.1.16, and EG.5.1 variants. The results showed similar neutralization titers between XBB.1.16 and EG.5.1 and were lower compared to XBB.1.5. Sera from the same individuals boosted with the bivalent mRNA vaccine contained higher neutralizing antibody titers, providing a better cross-protection against Omicron XBB.1.5, XBB.1.16 and EG.5.1 variants. Previous history of infection during the epidemiological waves of BA.1/BA.2 and BA.4/BA.5, poorly enhanced neutralization activity of serum samples against XBBs and EG.5.1 variants. Our data highlight the continued immune evasion of recent Omicron subvariants and support the booster administration of BA.4/5 bivalent vaccine, as a continuous strategy of updating future vaccine booster doses to match newly emerged SARS-CoV-2 variants.

## 1. Introduction

The continued evolution and emergence of new variants of severe acute respiratory syndrome coronavirus 2 (SARS-CoV-2) have been responsible for successive global waves of infection. Omicron has become the dominant variant since its first report in late November 2021 in South Africa [1]. Due to its high transmissibility [2] and immune evasion [3], Omicron successfully replaced other variants, such as Delta, with many Omicron sublineages emerging over time [4]. Recently, three emergent Omicron subvariants, XBB.1.5 (Kraken), XBB.1.16 (Arcturus) and EG.5.1 (Eris), have expanded rapidly worldwide [5] (Figure 1A), and have been recognized as variants of interest (VOI) by WHO. In late February 2023, XBB.1.5 sublineages harboring the F486P substitution in the spike protein predominated worldwide [6] (Figure 1A). Then, XBB.1.16, emerged independently of XBB.1.5 (Figure 1B), also harboring the F486P substitution and two other new substitutions in the spike protein: E180V in the N-terminal domain (NTD) and T478R in the receptor-binding domain (RBD) [7]. EG.5.1 evolved from Omicron XBB.1.9 (another BA.2 descendant) and harbors two additional substitutions, the Q52H in the NTD of the spike and the F456L in the RBD of the spike relative to XBB.1.16 (Figure 1B,C) [8].

Older adults (aged > 60 years) are a high-risk group vulnerable to severe disease and death after SARS-CoV-2 infection [9], thus it has been demonstrated that there is a less robust immune response after COVID-19 vaccination in this priority group [10,11]. In that way, since SARS-CoV-2 spike mutations deeply contribute to immune escape and transmissibility, a major goal of vaccine design is the evaluation of an optimal protection for the elderly. Furthermore, the kinetics of neutralizing activity in sera elicited by vaccination and the capacity of boosters to enhance cross-protection against new SARS-CoV-2 variants in elderly cohorts are of great interest. With the emergence and rapid dispersion of Omicron BA.4/5 worldwide [1], it was shown that individuals vaccinated with four doses of prototype mRNA vaccine did not produce a strong immune response against these Omicron sublineages, highlighting the need for bivalent vaccines development targeting the prototype and the BA.4/5 spike protein [12,13]. Because the recently emerged Omicron subvariants have accumulated additional spike mutations, it is important to access the vaccine-elicited neutralization against these new sublineages. Herein we investigated the escape of XBB.1.5, XBB.1.16, and EG.5.1 Omicron sublineages from neutralizing antibodies in serum samples collected from a cohort of 59 volunteers aged between 62 and 97 years (median 73 years) after 1 and 4 months of 4th dose vaccination with ChAdOx1-S vaccine and after 3 months post- 5th dose with Comirnaty Bivalent Original/Omicron BA.4/BA.5 vaccine.

**Figure 1 vaccines-12-00144-f001:**
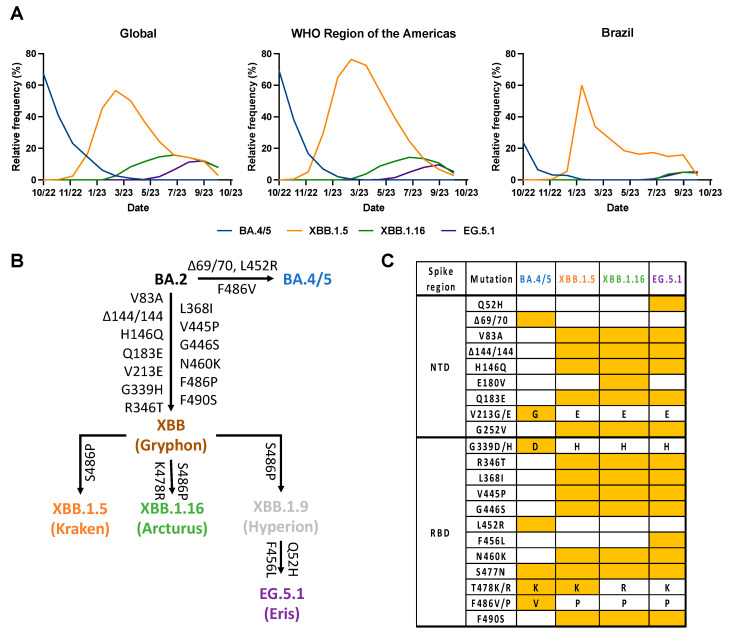
Proportion, evolution, and spike differences between BA.4/5, XBB (XBB.1.5, XBB.1.16, XBB.1.9) and EG.5.1 (Eris) SARS-CoV-2 variants. (**A**) Relative frequency (%) of BA.4/5, XBB (XBB.1.5, XBB.1.16, XBB.1.9) and EG.5.1 (Eris) globally (right), in the Americas (center), and Brazil (left) from October 2022 to October 2023. Data were collected in GISAID [5]. (**B**) Spike convergent evolution chart with BA.2, BA.4/5, XBB (XBB.1.5, XBB.1.16, XBB.1.9), and EG.5.1 (Eris). Arrows denote direct relationships between variants with the corresponding spike mutations written along them. (**C**) Comparison of BA.4/5, XBB.1.5, XBB.1.16, and EG.5.1 Omicron sublineages’ spike region. Amino acid mutations and deletions (Δ) are indicated in reference to the Wuhan-1 (614D) strain. Only distinct mutations between these variants were represented. NTD, N-terminal domain of the spike glycoprotein; RBD, receptor binding domain of the spike glycoprotein. Mutations were determined by the Outbreak.info genomic reports website [14].

## 2. Materials and Methods

### 2.1. Ethical Statement

We conducted a prospective cohort study of older adults (age > 60 years) receiving the pandemic COVID-19 vaccine between November 2022 and June 2023 at Dom Pedro II Geriatric and Convalescent Hospital, Irmandade da Santa Casa de Misericórdia de São Paulo, in São Paulo, Brazil. All subjects or a legally responsible person provided written informed consent before inclusion in the study, which was approved by the institutional ethics committee from the University of São Paulo and Dom Pedro II Geriatric and Convalescent Hospital (CEPSH-ICB/USP; CAAE: 66951221.2.0000.5467 and 69479523.2.0000.5467). The inclusion criteria were willingness to attend scheduled blood sampling visits and being vaccinated with all COVID-19 doses. The exclusion criteria were a history of anaphylaxis or hypersensitivity to vaccines and no vaccination record. The study was conducted in compliance with all International Council for Harmonisation Good Clinical Practice guidelines and the ethical principles of the Declaration of Helsinki. All SARS-CoV-2 experiments were performed in a Biosafety Level 3 (BSL-3) laboratory in the Institute of Biomedical Sciences (ICB) at the University of São Paulo (USP), São Paulo, Brazil. No diagnosis or treatment was involved.

### 2.2. Study Cohort and Sample Collection

Individuals included in this study have received five doses of the Coronavirus Disease 2019 (COVID-19) vaccine, following the regimen of two initial doses of ChAdOx1-S (Oxford/AstraZeneca) 4 weeks apart, followed by a booster with BNT162b2 (Pfizer-BioNTech) after 7 months post-primary vaccination. A fourth dose (second booster) of ChAdOx1-S was administered 11 months after the 3rd dose. Finally, five months after the 3rd dose, the volunteers were boosted with the 5th dose (third booster) of Comirnaty Bivalent Original/Omicron BA.4/BA.5 vaccine (Pfizer (New York, NY, USA)-BioNTech (Mainz, Germany)). Full cohort and demographic information are provided in Appendix A. Blood samples were taken at three timepoints: 1 (median 36 days) and 4 months (median 126 days) post-4th dose with ChAdOx1-S vaccine, and after 3 months (median 91 days) post-5th dose with Comirnaty Bivalent Original/Omicron BA.4/BA.5 vaccine (Appendix A). The serum was obtained by centrifuging the whole blood for 10 min at 4 °C and 2000 r.p.m. and stored at −20 °C until further use.

### 2.3. Cell and Viruses

Vero CCL-81 cells were purchased from the American Type Culture Collection (ATCC), and were maintained in DMEM (#D0069, Vitrocell, Nova Campinas, SP, Brazil) supplemented with 10% FBS (#S0011, Vitrocell) and 1% penicillin-streptomycin (#15140122, Thermo Fisher Scientific, Waltham, MA, USA) at 37 °C with 5% CO2. Vero CCL-81 cell line was tested negative for Mycoplasma before use, by conventional PCR, as previously described by Wong-Lee and Lovett, 1993 [15]. The ancestral B.1.1.28 (D614G) [16], B.1.1.529 (Omicron) sublineages BA.4, XBB.1.5, XBB.1.16, and EG.5.1 were isolated from NP samples and NGS sequenced. See the Data Availability Statement for the complete sequence information, including GISAID’s access codes. All viruses were passaged only once in Vero CCL-81 cells, collected 3 d.p.i., aliquoted and stored at −80 °C for subsequent experiments.

### 2.4. Cytopathic Effect-Based Virus Neutralization Test (CPE-VNT)

The CPE-VNT was performed following the protocol described by Wendel et al., 2020 [17]. Briefly, 5 × 104 cells/mL of Vero cells (ATCC CCL-81) were seeded 24 h before the infection in a 96-well plate. Serum samples were initially inactivated for 30 min at 56 °C. We used 8 dilutions (two-fold) of each serum (1:20 to 1:2560). Subsequently, the serum was mixed vol/vol with 10^3^ TCID_50_/mL of each SARS-CoV-2 variant and pre-incubated at 37 °C for 1 h to allow virus neutralization. Then, the serum plus virus mixture was transferred onto the confluent cell monolayer and incubated for 3 days at 37 °C, under 5% CO_2_. After 72 h, the plates were analyzed directly with the automated microscopy EVOS™ M5000 Imaging System (#AMF5000, Invitrogen^TM^, Waltham, MA, USA). In each assay, a convalescent serum sample was used as a positive control [18] and as a negative pre-pandemic serum sample (collected in 2017) [19]. The sample dilution resulting in 50% of virus neutralization (NT_50_) was calculated using the method of Spearman and Karber [20,21]. Each sample was carried out in triplicate for the NT_50_ calculation. Samples with no neutralization and observed CPE from the first dilution (1:20) were considered to have VNT_100_ titers equal to 10, which is equivalent to half of the limit of detection (LOD).

### 2.5. Statistical Analysis and Reproducibility

Categorical variables were summarized as number (n) and percentage (%) and continuous variables as the geometric mean with 95% confidence intervals (95%CI) or median with interquartile ranges (IQRs). For plot purposes, the geometric mean titer (GMT) calculations, statistical analysis, and sera with undetectable (NT_50_ < 18) nAbs titers were assigned an NT_50_ of 14 (representing triplicates with a VNT_100_ titer equal to 10). Comparison between neutralization titers was performed using a Wilcoxon matched-pairs signed-rank test or a nonparametric Mann–Whitney test, when applicable, using GraphPad Prism version 10.0.3 for macOS (GraphPad Software, Boston, MA, USA, www.graphpad.com). Absolute *p* values are provided in Appendix A. *p* < 0.05 was considered statistically significant. For the infection history analyses, two epidemiological Omicron wave periods in the São Paulo state, Brazil, were defined as follows: 1st Omicron wave by BA.1/BA.2 (from January 2022 to May 2022) and 2nd Omicron wave by BA.4/BA.5 subvariants (from June 2022 to January 2023), as previously described by Hojo-Souza et al., 2023 [22] and based on the Fiocruz Genomics Network (https://www.genomahcov.fiocruz.br/dashboard-en/, accessed on 20 October 2023). Images were assembled using Adobe Illustrator, version 28.0. No statistical method was used to predetermine the sample size since the samples were collected based on the availability of volunteers in the Dom Pedro II Geriatric and Convalescent Hospital. No data were excluded from the analyses.

## 3. Results

### 3.1. Study Population Characteristics

A total of 59 older adults were enrolled in this study. The patients had a median age of 73 years (IQR 67–81), of which 42.4% (n = 25) were white and 26 (44.1%) were female (Table 1 and Appendix A). Of all the participants, 62.7% (n = 37) reported being RT-qPCR positive for SARS-CoV-2. The days between the most recent positive SARS-CoV-2 PCR test and the 1st blood collection varied from 121 to 597 days, with a median of 281 days (95% confidence interval [CI] 142–290; Appendix A). In general, the patients had few comorbidities, with high rates of dementia and dysmobility syndrome (Table 1).

### 3.2. Neutralization against Omicron Sublineages after 4th Dose with Oxford/AstraZeneca (ChAdOx1-S) Vaccine

Sera collected 1 month (median 36 days) after the 4th dose of the Oxford/AstraZeneca (ChAdOx1-S) vaccine (Figure 2A) neutralized D614G (BRA-HIAE02/2020), BA.4/5, XBB.1.5, XBB.1.16, and EG.5.1 SARS-CoV-2 variants with geometric mean titers (GMTs) of 1488, 803, 122, 56, and 81, respectively (Figure 2B and Appendix A). The neutralizing GMTs against BA.4/5, XBB.1.5, XBB.1.16, and EG.5.1 viruses were 1.8-fold (*p* = 0.6089), 12.2-fold (*p* < 0.0001), 26.6-fold (*p* < 0.0001) and 18.4-fold (*p* < 0.0001) lower than the GMT against the parental D614G virus, respectively (Figure 1B). XBB.1.16 showed a 2.3-fold reduction (*p* = 0.0006) when compared with XBB.1.5, and EG.5.1 showed no significant reduction (*p* = 0.157) when compared with XBB.1.16. Individuals boosted developed a measurable nAb response against D614G, BA.4/5, XBB.1.5, XBB.1.16, and EG.5.1 SARS-CoV-2 variants at rates of 100% (n = 59/59), 100% (n = 59/59), 92% (n = 54/59), 83% (n = 44/59) and 92% (n = 54/59), respectively (Figure 2B and Appendix A).

Sera collected 4 months (median 126 days) after the 4th dose of the Oxford/AstraZeneca (ChAdOx1-S) vaccine neutralized D614G, BA.4/5, XBB.1.5, XBB.1.16, and EG.5.1 SARS-CoV-2 variants with geometric mean titers (GMTs) of 1191, 430, 72, 31, and 41, respectively (Figure 2C and Appendix A). The neutralizing GMTs against BA.4/5, XBB.1.5, XBB.1.16, and EG.5.1 viruses were 2.8-fold (*p* = 0.0187), 16.5-fold (*p* < 0.0001), 38.4-fold (*p* < 0.0001) and 29.0-fold (*p* < 0.0001) lower than the GMT against the parental D614G virus, respectively (Figure 2E). XBB.1.16 showed a 2.2-fold reduction (*p* = 0.0016) when compared with XBB.1.5. A lower proportion of seroconversion for XBB.1.5, XBB.1.16, and EG.5.1 was observed at this time point post-vaccination at rates of 86% (n = 51/59), 70% (n = 41/59), and 80% (n = 47/59), respectively (Figure 2C and Appendix A). No difference was observed in the seroconversion against the D614G and BA.4/5 variants. These results indicate that (1) XBB-lineage subvariants, including XBB.1.5, XBB.1.16, and EG.5.1, demonstrated marked reductions in antibody neutralization relative to D614G and BA.4/5 (Figure 2A,B). (2) EG.5.1 exhibited modestly decreased neutralization relative to XBB.1.5 and XBB.1.16 (*p* > 0.05), and 4 mo. post 4th dose with Oxford/AstraZeneca (ChAdOx1-S), the neutralizing activity against the newly emerged Omicron sublineages was reduced, and (3) the rank of neutralization evasion was in the order of BA.4/5 < XBB.1.5 < XBB.1.16 = EG.5.1.

### 3.3. Neutralization against Omicron Sublineages after the 5th Dose with BA.4/5 Bivalent Vaccine (Pfizer-BioNTech)

BA.4/5 bivalent sera, collected at 3 months (median 91 days) after the boost, neutralized D614G, BA.4/5, XBB.1.5, XBB.1.16, and EG.5.1 SARS-CoV-2 variants with GMTs of 1958, 1454, 328, 203, and 208, respectively (Figure 2D and Appendix A). The neutralizing GMTs against BA.4/5, XBB.1.5, XBB.1.16, and EG.5.1 viruses were 1.3-fold (*p* = 0.0187), 6.0-fold (*p* < 0.0001), 9.6-fold (*p* < 0.0001), and 9.4-fold (*p* < 0.0001) lower than the GMT against the parental D614G virus, respectively (Figure 2D). Notably, neutralizing antibody titers against EG.5.1 were markedly less than those against XBB.1.5, with a 1.6-fold reduction (*p* = 0.0328), but not against XBB.1.16 (*p* > 0.99). The data indicate that the BA.4/5 bivalent booster elicits high neutralizing titers against D614G and BA.4/5 variants and confers moderate cross-protection against XBB.1.5, XBB.1.16, and EG.5.1, measured 3 months after the boost.

Our data also showed a slight decrease in the measured neutralizing antibodies from 1 to 4 months post-4th dose with ChAdOx1-S, but were not statistically significant (*p* > 0.05) unless against BA.4/5 (*p* = 0.0185; Figure 2E). Remarkably, a significant increase in the nAbs titers was observed at 3 months post-bivalent booster when compared to titers after 4 months of ChAdOx1-S vaccination against D614G (1.6-fold; *p* = 0.0358), BA.4/5 (3.3-fold; *p* < 0.0001), XBB.1.5 (4.2-fold; *p* < 0.0001), XBB.1.16 (5.9-fold; *p* < 0.0001), and EG.5.1 (5.4-fold; *p* < 0.0001) viruses (Figure 2E). A statistically significant difference against XBB.1.5 (2.7-fold; *p* < 0.0001), XBB.1.16 (3.6-fold; *p* < 0.0001), and EG.5.1 (2.7-fold; *p* < 0.0001) viruses (Figure 2E) was also observed when comparing the titers 1 month post-vaccination with ChAdOx1-S and 3 months post-bivalent booster. These results suggest that vaccination with the Comirnaty Bivalent Original/Omicron BA.4/BA.5 vaccine produced a more robust neutralizing response against the selected variants in older adults.

### 3.4. Influence of History of Infection by SARS-CoV-2 in nAbs Titers

We further investigated if older adults with a history of SARS-CoV-2 infection had higher neutralizing titers against the studied variants at the different time points of sample collection. We observed slightly higher titers in individuals with infection history but these were not statistically significant (*p* > 0.05) unless against BA.4/5 (GMTs:279 vs. 556, *p* = 0.0367) and EG.5.1 (GMTs: 31 vs. 49, *p* = 0.0346) at 4 months post-vaccination with the ChAdOx1-S vaccine (Figure 3A,B and Appendix A). When analyzing the samples with infection history and separating them into two groups, those with a previous infection before the epidemiological wave of BA.4/BA.5 (until June 2022, mainly during the BA.1/BA.2 wave) and those during the BA.4/5 wave (from June 2022 to January 2023), it was noticed that the difference against BA.4/5 (GMTs: 279 vs. 823, *p* = 0.0109) and EG.5.1 (GMTs: 31 vs. 55, *p* = 0.0328) was related to samples with infection history during the BA.4/5 wave (Figure 3A,B and Appendix A). These results suggest that BA.1/BA.2 or BA.4/BA.5 breakthrough infection did not substantially increase the magnitude of nAbs’ levels against the tested Omicron sublineages XBB.1.5, XBB.1.16, and EG.5.1.

No difference was observed at 3 months post vaccination with the bivalent vaccine between patients without infection history, with a previous infection before or during the epidemiological wave of BA.4/BA.5, respectively, against D614G (GMTs: 2305 vs. 1789 vs. 1762, *p* > 0.05), BA.4/5 (GMTs: 1498 vs. 1470 vs. 1379, *p* > 0.05), XBB.1.5 (GMTs: 279 vs. 447 vs. 281, *p* > 0.05), XBB.1.16 (GMTs: 189 vs. 246 vs. 177, *p* > 0.05), and EG.5.1 (GMTs: 178 vs. 245 vs. 209, *p* > 0.05), as shown in Appendix A.

## 4. Discussion

The immune response in older adults after the COVID-19 vaccination remains understudied. In this study, we measured the levels of neutralizing antibodies (nAbs) against D614G, BA.4/5, XBB.1.5, XBB.1.16, and EG.5.1 viruses in a Brazilian cohort of older adults after vaccination with a 4th dose of the ChAdOx1-S vaccine and a 5th dose of the bivalent mRNA vaccine. We found higher levels of nAbs after the 5th dose and a substantial immune escape in sera from these individuals against newly emerged SARS-CoV-2 variants.

Vaccination with ChAdOx1 has been demonstrated to protect older adults [23,24]. However, with the rapidly evolving of SARS-CoV-2, and mainly several Omicron sublineages (i.e., XBB) that exhibit enhanced fusogenicity and substantial immune evasion in the elderly population [25], the administration of a booster dose with mRNA bivalent vaccines targeting the ancestral and the BA.4/5 spike protein was recommended to generate immunity. Our data on Omicron neutralization following a 4th dose of ChAdOx1-S vaccine mirrors others who showed decreases in neutralization of >35-fold against XBB.1.5, XBB.1.16, and EG.5.1 [26,27]. This difference is reduced by less than 10-fold against the same variants after 3 months of a booster shot with Comirnaty Bivalent Original/Omicron BA.4/BA.5 vaccine, as observed by others [27,28]. In that way, it is clear from our data that boosting with a bivalent mRNA vaccine generates a much higher overall titer of neutralizing antibodies, enhancing cross-protection against high immune evasive variants such as XBB.1.5, XBB.1.16, and EG.5.1 [29].

Recent studies have demonstrated that vaccinated individuals with SARS-CoV-2 infection history develop a stronger humoral immune response against Omicron subvariants [12,30]. However, the impact of the infection subvariant and the time between infection and sample collection might influence these observations. Here we showed that older adults with reported infection mainly during the first Omicron wave (BA.1/BA.2, from January 2022 to May 2022, São Paulo, Brazil) [22,31] did not present higher levels of nAbs against D614G, BA.4/5, XBB.1.5, XBB.1.16, or EG.5.1. Furthermore, individuals infected during or after the second Omicron wave (BA.4/5, from June 2022, São Paulo, Brazil) showed discretely higher titers against only BA.4/5 and EG.5.1 (*p* < 0.05). In the beginning of the Omicron era, a low cross-neutralization against Omicron from previously non-Omicron infection was proposed [32], reinforcing the importance of vaccination to mitigate Omicron dispersion. In that way, our data is similar to Yang et al., 2023, who observed low levels of neutralizing antibodies against XBB.1.5 after BA.5 infection [33]. Also, it has been shown that vaccinated individuals with BA.1, BA.2, BA.5, or BF.7 breakthrough infections had a low humoral response against XBB lineages [34,35].

Some limitations of this study warrant consideration. First, we have not examined either the binding antibody titers against spike and/or NP or the cellular immune response. Neutralizing antibodies together with the T-cell immune response protect patients from severe disease and death [36,37] and can provide different insights when analyzed together. Second, we were not able to investigate possible differences in other vaccine regimens since all the participants took the same vaccine doses. Third, while this study, to our knowledge, represents one of the largest cohort studies to date on the antibody response in older adults after COVID-19 vaccination, it still represents a relatively small sample size. Fourth, we were not able to collect blood samples just before the boost with the mRNA bivalent dose; in that way, we were not able to determine the nAbs baseline titers. Finally, we did not perform the sequencing of samples from the participants with infection history, because of sample unavailability.

## 5. Conclusions

In conclusion, our neutralization data support that newly emerged Omicron sublineages are still enhancing their immune escape from vaccine-elicited neutralization. Among tested Omicron sublineages, XBB.1.16 and EG.5.1 show the greatest resistance to vaccine-induced neutralization, implying a possible reason why these new sublineages replaced others in circulation and became dominant in the past. Furthermore, our findings suggest a vaccination update strategy that future boosters ought to reflect new circulating SARS-CoV-2 VOIs, in line with the recent real-world effectiveness of the BA.5 bivalent booster [38], and highlights the importance of the currently licensed XBB.1.5 vaccine administration as soon as available, independent of history of infection with SARS-CoV-2.

## Figures and Tables

**Figure 2 vaccines-12-00144-f002:**
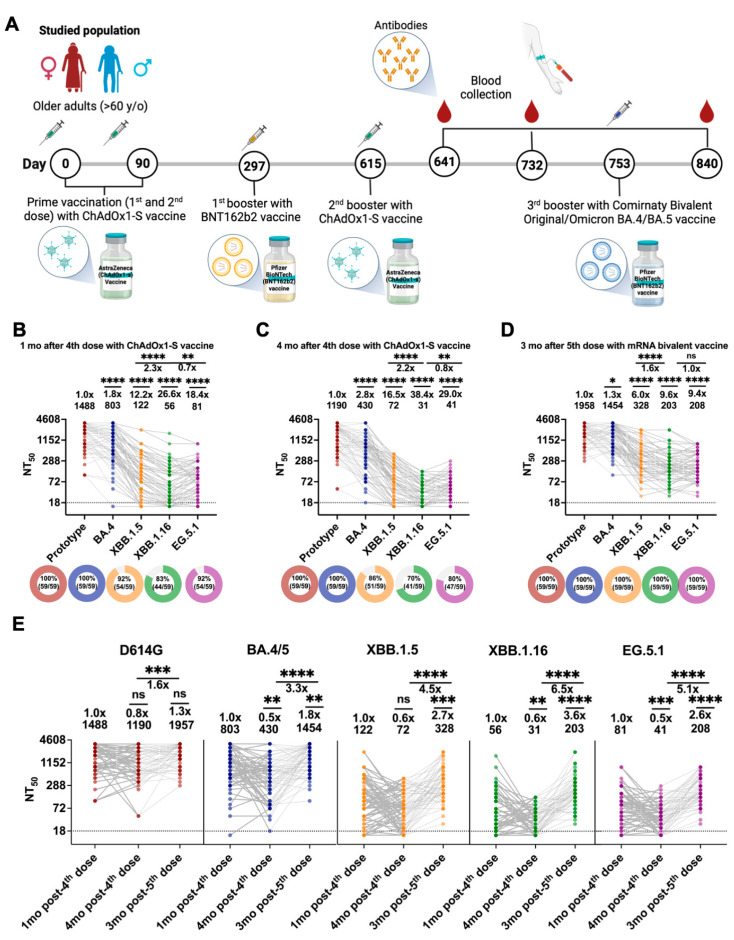
Neutralization against Omicron sublineages BA.4/5, XBB.1.5, XBB.1.16, and EG.5.1. (**A**) Study design. (**B**–**D**) NT_50_ values of 59 human sera against Omicron BA.4/5, XBB.1.5, XBB.1.16, and EG.5.1, collected after (**B**) 1 month (median 36 days), (**C**) 4 months (median 126 days) post-4th dose with parental ChAdOx1-S vaccine, and (**D**) 3 months (median 86 days) post-5th dose with bivalent mRNA vaccine. (**E**) Paired analysis of 1 and 4 months post 4th dose with parental ChAdOx1-S vaccine, and 3 months post 5th dose with bivalent mRNA vaccine serum neutralizing titers against D614G variant, Omicron BA.4/5, XBB.1.5, XBB.1.16, and EG.5.1 from 59 individuals. (**B**–**D**) Colors designate SARS-CoV-2 variants, data points represent individual subjects and dotted lines represent the lower limit of detection of NT_50_ (LOD = 18). Geometric mean titers (GMTs) are noted above the respective groups. The fold of GMT reduction against each Omicron sublineage compared with the GMT against B.1.1.28 (SP-HIAE-ID02/2020) are noted above the respective GMTs. Groups were compared by Wilcoxon matched-pairs signed-rank test (ns = not significant, * = *p* < 0.05, ** = *p* < 0.01, *** = *p* < 0.001, **** = *p* < 0.0001). Panel (**A**) was created with BioRender.com (accessed on 10 November 2023) and panels (**B**–**E**) were generated using GraphPad Prism v10.0 software. See Appendix A for complete data.

**Figure 3 vaccines-12-00144-f003:**
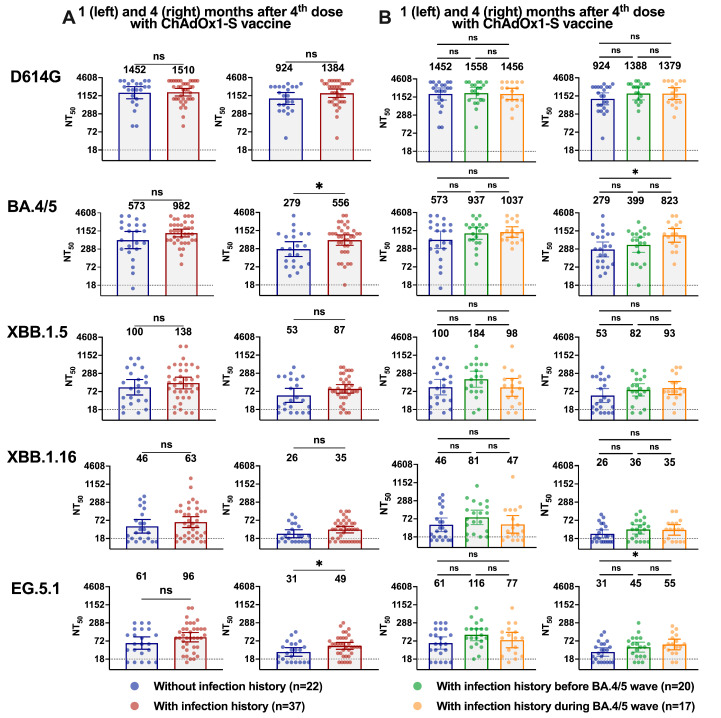
Neutralization against Omicron sublineages BA.4/5, XBB.1.5, XBB.1.16, and EG.5.1 in individuals with or without a history of infection by SARS-CoV-2. (**A**) NT_50_ values of sera from older adults without (n = 22) and with (n = 37) infection history, against D614G, Omicron BA.4/5, XBB.1.5, XBB.1.16, and EG.5.1, collected after (**A**) 1 month (median 36 days; **left** column) and (**B**) 4 months (median 126 days; **right** column) post-4th dose with parental ChAdOx1-S vaccine. (**B**) NT_50_ values of sera from individuals with infection history were subdivided into two groups: the ones infected before the BA.4/5 wave (mainly during the BA.1/BA.2 wave) and the ones infected during the BA.4/5 wave. Geometric mean titers (GMTs) are noted above the respective groups. Colors designate infection history, data points represent individual subjects, bars represent GMT with 95% confidence intervals (CI) and dotted lines represent the lower limit of detection of NT_50_ (LOD = 18). Groups were compared by the non-parametric Mann–Whitney test (ns = not significant, * = *p* < 0.05). Panels (**A**,**B**) were generated using GraphPad Prism v10.0 software. See Appendix A for the complete data.

**Table 1 vaccines-12-00144-t001:** Demographic and clinical characteristics of the older adults’ cohort (n = 59). See Appendix A for the complete data.

Characteristics	Cohort (n = 59)
Age, years—median (IQR)	73 (67–81)
Sex	
Female—n (%)	26 (44.1%)
Male—n (%)	33 (59.1%)
Race	
Asian	2 (3.4%)
Black	15 (25.4%)
Mixed	17 (28.8%)
White	25 (42.4%)
Reported SARS-CoV-2 infection by RT-qPCR	
Positive	37 (62.7%)
Negative	22 (37.3%)
Comorbidities (n, %)	
Alzheimer’s disease	1 (1.7%)
Cerebral palsy	1 (1.7%)
Dementia	14 (23.7%)
Dysmobility syndrome	13 (22.0%)
Hypertension	9 (15.2%)
Leprosy	1 (1.7%)
Rheumatic fever	1 (1.7%)
Schizophrenia	4 (6.8%)
Sequalae of Traumatic Brain Injury (TBI) or stroke	10 (16.9%)
Type 2 diabetes mellitus (T2DM)	2 (3.4%)
None	4 (6.8%)

## Data Availability

The raw data that support the findings of this study are shown in Appendix A. The sequence of SARS-CoV-2 variants can be accessed through GISAID (https://gisaid.org) with the following codes: ancestral B.1.1.28—D614G (hCoV-19/Brazil/SP-HIAE-ID02/2020, EPI_ISL_861868), B.1.1.529 (Omicron) sublineages BA.4 (hCoV-19/Brazil/SP-LVCM-061/2022, EPI_ISL_17959808), XBB.1.5 (hCoV-19/Brazil/SP-LVCM-045/2023, EPI_ISL_17959794), XBB.1.16 (hCoV-19/Brazil/SP-HIAE-ID2720/2023, EPI_ISL_17505614) and EG.5.1 (hCoV-19/Brazil/SP-HIAE-ID2826/2023, EPI_ISL_18128356). Any other information is available upon request.

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
