# Peer review of "Immune Evasion of SARS-CoV-2 Omicron Subvariants XBB.1.5, XBB.1.16 and EG.5.1 in a Cohort of Older Adults after ChAdOx1-S Vaccination and BA.4/5 Bivalent Booster"

_vaccines, 2024, doi:10.3390/vaccines12020144_

Round 1
Reviewer 1 Report
Comments and Suggestions for Authors This work addresses an important topic related to the immune response to SARS-CoV-2 vaccines in older adults. However, the study has multiple limitations. Specific comments are as follows:- What was the rationale to analyze that specific mixed vaccination strategy?
- There is a very limited amount of information provided by the results in this manuscript.
- Figure 2D. Was the difference between the prototype and XBB.1.16 and EG.5.1 statistically different?
- Figure 3. How was the analysis of infection history of the samples after the 5 vaccine doses?
- This study must show additional analysis on the neutralizing effect against the Spike protein.
- Which vaccine booster was more effective to induce a better neutralizing response in older adults?
- How was the disease outcome of the analyzed individuals?
Comments on the Quality of English Language
The manuscript is well written for most part. However, it will benefit from an additional round of editing.
Author Response
Response to Reviewer 1 Comments
Thank you very much for taking the time to review our manuscript. Please find the detailed responses below and the corresponding revisions/corrections highlighted/in track changes in the re-submitted files.
Point-by-point response to Comments and Suggestions for Authors
Overall comment: This work addresses an important topic related to the immune response to SARS-CoV-2 vaccines in older adults. However, the study has multiple limitations.
Response: We thank the reviewer for the comment and we recognise all the limitations of this study, as described in the discussion section (lines 645-656).
Comment 1: What was the rationale to analyze that specific mixed vaccination strategy?
Response 1: We thank the reviewer for the question. Since the participants of the study live in a public nursing home in the city of São Paulo, Brazil, they were vaccinated in accordance with the recommendations of the Brazilian Ministry of Health and the availability of vaccines in the city. No other criteria were taken into consideration, and no choices regarding different vaccines were made.
Comment 2: There is a very limited amount of information provided by the results in this manuscript.
Response 2: We thank the reviewer for the comment, we have reviewed the entire results section and included extra information (lines 469-492 and 533-538).
Comment 3: Figure 2D. Was the difference between the prototype and XBB.1.16 and EG.5.1 statistically different?
Response 3: We thank the reviewer for the question. We have described this differences in lines 458-465, as showed in Figure 2D (upper bars): "The neutralizing GMTs against BA.4/5, XBB.1.5, XBB.1.16 and EG.5.1 viruses were 1.3-fold (p=0.0187), 6.0-fold (p<0.0001), 9.6-fold (p<0.0001) and 9.4-fold (p<0.0001) lower than the GMT against the parental D614G virus, respectively (Figure 1F). Notably, neutralizing antibody titers against EG.5.1 were markedly less than those against XBB.1.5, with a 1.6-fold reduction (p=0.0328), but not against XBB.1.16 (p>0.99)".
Comment 4: Figure 3. How was the analysis of infection history of the samples after the 5 vaccine doses?
Response 4: We thank the reviewer for the question. The infection history of the participants were accessed through medical records. Furthermore, the Laboratory of Clinical and Molecular Virology, coordinated by Prof. Dr. Edison L. Durigon, is responsible for the COIVID-19 molecular diagnosis of the patients from the Dom Pedro II Geriatric and Convalescent Hospital, so we were able to track the entire infection history of the participants of this study. In that way, no infections were reported between the time they took the 5th dose and the last blood collection (Tables S1 and S2).
Comment 5: This study must show additional analysis on the neutralizing effect against the Spike protein.
Response 5: Thank you for your comment. Unfortunately we were not able to access the binding activity against the specific spikes protein of the investigated variants, however we strongly believe that our neutralization data represent the impact of mutations in the spike protein, mainly in the RBD region of the selected variants. We have included this limitation in the discussion section (lines 645-656).
Comment 6: Which vaccine booster was more effective to induce a better neutralizing response in older adults?
Response 6: We thank the reviewer for the question. We are not able to properly answer that question, since we are not comparing two groups that received different vaccines at the same time-point. Actually we are analysing the same patients after the admistration of a 4th dose with ChAdOx1-S and 5th dose with the bivalent vaccine dose. Taking that in account, we observed a higher neutralisation titer against the selected variants after vaccination with Comirnaty Bivalent Original/Omicron BA.4/BA.5 vaccine (Pfizer-BioNTech). We have included a sentence in lines 469-492, regarding that "Remarkably, a significant increase in the nAbs titers was observed at 3 months post-bivalent booster when compared to titers after 4 months of ChAdOx1-S vaccination against D614G (1.6-fold; p=0.0358), BA.4/5 (3.3-fold; p<0.0001), XBB.1.5 (4.2-fold; p<0.0001), XBB.1.16 (5.9-fold; p<0.0001) and EG.5.1 (5.4-fold; p<0.0001) viruses (Figure 2E). A statistically significant difference against XBB.1.5 (2.7-fold; p<0.0001), XBB.1.16 (3.6-fold; p<0.0001) and EG.5.1 (2.7-fold; p<0.0001) viruses (Figure 2E) was also observed when comparing the titers 1 month post-vaccination with ChAdOx1-S and 3 months post-bivalent booster. These results suggest that vaccination with the Comirnaty Bivalent Original/Omicron BA.4/BA.5 vaccine produced a more robust neutralizing response against the selected variants in older adults".
Comment 7: How was the disease outcome of the analyzed individuals?
Response 7: Thank you for your question. We have included the symptomatology of the participants after SARS-CoV-2 infection (the ones with infection history) in Table S1. None of them need hospitalisation or died after the SARS-CoV-2 infection.
Response to Comments on the Quality of English Language
Point 1: The manuscript is well written for most part. However, it will benefit from an additional round of editing.
Response 1: We thank the reviewer for the feedback, we have revised the entire manuscript and made some edits to increase the quality of the spelling.
Reviewer 2 Report
Comments and Suggestions for Authors
The manuscript by Machado et al described the neutralizing activities of sera samples from older people after vaccination with ChAdOx1-S and Comirnaty Bivalent Original/Omicron BA.4/BA.5. The results exhibited the continued immune evasion of new Omicron subvariants and well supported the opinion that future boosters should match newly emerged circulating SARS-CoV-2 variants. The manuscript is well prepared, and the results are well presented.
My minor comments are as follows:
1. The full name of any abbreviation should be provided when it first appears, for example, GMT. And the full name will not be necessary after its first appearance to keep a concise manuscript.
2. The race row of Table 2, numbers in brackets lack %.
Comments on the Quality of English LanguageThe quality of English language is acceptable.
Author Response
Response to Reviewer 2 Comments
Thank you very much for taking the time to review our manuscript. Please find the detailed responses below and the corresponding revisions/corrections highlighted/in track changes in the re-submitted files.
Point-by-point response to Comments and Suggestions for Authors
Overall comment: The manuscript by Machado et al described the neutralizing activities of sera samples from older people after vaccination with ChAdOx1-S and Comirnaty Bivalent Original/Omicron BA.4/BA.5. The results exhibited the continued immune evasion of new Omicron subvariants and well supported the opinion that future boosters should match newly emerged circulating SARS-CoV-2 variants. The manuscript is well prepared, and the results are well presente
Response: We thank the reviewer for the positive feedback regarding our study and we appreciate the time spent to review it.
Comment 1: The full name of any abbreviation should be provided when it first appears, for example, GMT. And the full name will not be necessary after its first appearance to keep a concise manuscript.
Response 1: We thank the reviewer for the suggestion, we have corrected it through the entire manuscript. We provided the full name of each abbreviation only once, when it first appears, as GMT, NT50, LOD and nAbs.
Comment 2: The race row of Table 2, numbers in brackets lack %.
Response 2: We thank the reviewer for spotting this typo and have corrected as requested (Table 1, line 408).
Response to Comments on the Quality of English Language
Point 1: The quality of English language is acceptable.
Response 1: We thank the reviewer for the positive feedback. We have revised the entire manuscript and made some minor editing, aiming to improve the quality of the writing.
Reviewer 3 Report
Comments and Suggestions for Authors
The authors highlighted the continued problem of COVID immune evasion due to variants and how vaccine responses work against them. The novelty of this paper is the neutralization responses against multiple vaccinations of ChAdOx1-S in a high risk group followed by a mRNA boost. This work provides more information on an approved COVID vaccine against variants that is still in use but, is being displaced due to mRNA variant boosts. They authors covered cross neutralization and how the response may have been influenced due to previous infections and what virus it was. They size of the population is always a risk is science. It is unfortunate that this was such a small group as an overall vaccination study, but it is what was available to the authors. As they point out it is large study for this distinct risk group. The authors did cite flaws in their own research about sample collection and short coming in planning. The conclusions drawn are similar to what other vaccines are seeing, is that boosters with current strains provide better protection.
Line 65: correct "worldwide1"
Thank you for highlighting the shortfalls in the study starting at Line:301
Supplemental Table S2 Vaccination Records does not need to be provided as it does not add to the core research of the paper.
Author Response
Response to Reviewer 3 Comments
Thank you very much for taking the time to review our manuscript. Please find the detailed responses below and the corresponding revisions/corrections highlighted/in track changes in the re-submitted files.
Point-by-point response to Comments and Suggestions for Authors
The authors highlighted the continued problem of COVID immune evasion due to variants and how vaccine responses work against them. The novelty of this paper is the neutralization responses against multiple vaccinations of ChAdOx1-S in a high risk group followed by a mRNA boost. This work provides more information on an approved COVID vaccine against variants that is still in use but, is being displaced due to mRNA variant boosts. They authors covered cross neutralization and how the response may have been influenced due to previous infections and what virus it was. They size of the population is always a risk is science. It is unfortunate that this was such a small group as an overall vaccination study, but it is what was available to the authors. As they point out it is large study for this distinct risk group. The authors did cite flaws in their own research about sample collection and short coming in planning. The conclusions drawn are similar to what other vaccines are seeing, is that boosters with current strains provide better protection.
Response: We thank the reviewer for the positive feedback regarding our study and we appreciate the time spent to review it.
Comment 1: Line 65: correct "worldwide1"
Response 1: We thank the reviewer for spotting this typo and have corrected as requested (line 103).
Comment 2: Thank you for highlighting the shortfalls in the study starting at Line:301
Response 2: We thank the reviewer for the comment.
Comment 3: Supplemental Table S2 Vaccination Records does not need to be provided as it does not add to the core research of the paper.
Response 3: We thank the reviewer for the suggestion; however, we decided to keep it since it was requested by one of the other reviewers.
Response to Comments on the Quality of English Language
Point 1: English language fine. No issues detected
Response 1: We thank the reviewer for the positive feedback.
Reviewer 4 Report
Comments and Suggestions for Authors
Overall, this is a nice study of the immunity to SARS-CoV-2 in this population. I don't see any major issues with the study - it is performed well and data supports the conclusion. Certainly, it does not go into depth on why this may be happening- but I think this is a nice contribution to the data.
There are a few minor things:
1. In table 1: might be good to include "none" or "none known" in the co-morbitities section.
2. Figure 3 is quite confusing and overwhelmed with data. Might be good to split this into two figures. One with the red and blue bars and one with the further analysis of the green and yellow bars.
3. In figure 3: there are data points below the detection limit. Why is that? they should be normalised to the detection limit. I also, therefore, ask whether the stats were performed on data that was all corrected to the deletion limit or included "zero"s? If the latter, they are not valid as you cannot detect anything in the range 0-18.
Author Response
Response to Reviewer 4 Comments
Thank you very much for taking the time to review our manuscript. Please find the detailed responses below and the corresponding revisions/corrections highlighted/in track changes in the re-submitted files.
Point-by-point response to Comments and Suggestions for Authors
Overall, this is a nice study of the immunity to SARS-CoV-2 in this population. I don't see any major issues with the study - it is performed well and data supports the conclusion. Certainly, it does not go into depth on why this may be happening- but I think this is a nice contribution to the data.
Response: We thank the reviewer for the positive feedback regarding our study and we appreciate the time spent to review it.
Comment 1: In table 1: might be good to include "none" or "none known" in the co-morbitities section.
Response 1: We thank the reviewer for the suggestion, we have included in Table 1 as requested (line 408).
Comment 2: Figure 3 is quite confusing and overwhelmed with data. Might be good to split this into two figures. One with the red and blue bars and one with the further analysis of the green and yellow bars.
Response 2: We thank the reviewer for the comment and suggestion. We agree with your considerations and changed the display of the graphs in Figure 3. In that way, to improve the visibility of the data, we rearranged the graphs, so in Figure 3A we displayed only the first analysis (with or without infection history -blue and red dots/bars) and in letter B the second analysis regarding time of infection (green and orange dots/bars). We also changed the legend of the figure to better explain it and facilitate its comprehension.
Comment 3: In figure 3: there are data points below the detection limit. Why is that? they should be normalised to the detection limit. I also, therefore, ask whether the stats were performed on data that was all corrected to the deletion limit or included "zero"s? If the latter, they are not valid as you cannot detect anything in the range 0-18.
Response 3: We thank the reviewer for the comment and discussion. We have included in the methodology section the following statement in lines 259-261: "Samples with no neutralization and observed CPE from the first dilution (1:20) were considered to have VNT100 titers equal to 10, which is equivalent to half of the limit of detection (LOD)" and in lines 265-266 ''For plot purposes, geometric mean titer (GMT) calculations, and statistical analysis, sera with undetectable (NT50<18) nAbs titers were assigned an NT50 of 14 (representing triplicates with a VNT100 titer equal to 10)".
Response to Comments on the Quality of English Language
Point 1: English language fine. No issues detected
Response 1: We thank the reviewer for the positive feedback.